# BNT162b2 (Pfizer/BioNTech) COVID-19 Vaccination Was Not Associated with the Progression of Activity of the Exudative Form of Age-Related Macular Degeneration during Anti-VEGF Therapy

**DOI:** 10.3390/vaccines10111878

**Published:** 2022-11-07

**Authors:** Bernadetta Płatkowska-Adamska, Agnieszka Bociek, Joanna Krupińska, Magdalena Kal, Michał Biskup, Dorota Zarębska-Michaluk, Dominik Odrobina

**Affiliations:** 1Collegium Medicum, Jan Kochanowski University, 25-317 Kielce, Poland; 2Ophthalmology Clinic, Voivodeship Hospital, 25-727 Kielce, Poland; 3Hospital of the Ministry of the Interior and Administration, 25-375 Kielce, Poland; 4Department of Infectious Disease, Voivodeship Hospital, 25-317 Kielce, Poland; 5Ophthalmology Clinic, St. John Boni Fratres Lodziensis, 93-357 Łódź, Poland

**Keywords:** age-related macular degeneration, COVID-19, vaccine, BNT162b2

## Abstract

Background: Evaluation of the activity of the exudative form of age-related macular degeneration (AMD) during anti-vascular endothelial growth factor (anti-VEGF) therapy before and after administration of BNT162b2 (Pfizer/BioNTech) vaccination. Methods: The optical coherence tomography and best corrected visual acuity (BCVA) records of the two previous visits before the first dose of BNT162b2 (first pre-vaccination visit marked as “V-1”, the previous pre-vaccination “V-2”), and two subsequent visits after the second dose of vaccination (first visit after the second dose marked as “V1”, second visit after the second dose marked as “V2”) were collected for 63 eyes of 59 patients. Results: The difference in the average retinal thickness was observed between the last and each other checkpoint for the aflibercept group and in the overall outcome. The maximum thickness from the inner retinal surface to the inner border of RPE decreased during the observation; differences were observed comparing visits −2 and 1. Subretinal complex thickness decreased during follow-up, and the differences were observed between visits −2 and 2. There were no statistically significant differences in the BCVA and the occurrence of intraretinal cystoid fluid, serous PED, subretinal hyperreflective material, and retinal hemorrhage. Conclusion: In the present study, the activity of the exudative form of AMD did not deteriorate after the administration of the BNT162b2 vaccine.

## 1. Background

Since March 2020, the whole world, including Poland, has been struggling with a World Health Organization-declared pandemic caused by the severe acute respiratory syndrome coronavirus 2 (SARS-CoV-2). The infection leads to the development of a disease called coronavirus (COVID-19), associated with acute respiratory failure [1].

The first vaccinations against COVID-19 in Poland began on 26 December 2020, prioritizing the healthcare workers in the so-called ‘0’ group. The next stage, the National Vaccination Program in Poland, included people over 60, beginning with the oldest. Some of the first vaccines approved in Poland were BNT162b2 (Pfizer/BioNTech, New York, NY, USA), mRNA-1273 (Moderna Inc., Cambridge, MA, USA), ChAdOx1-S (Oxford/AstraZeneca, Cambridge, UK), and, from April 2021, Ad26.COV2.S (Johnson Pharm, NJ, USA). Primary protection with the above vaccines consisted of two doses given 28 days apart (28–94 days ChAdOx1-S) with the exception of the Ad26.COV2.S vaccine, whose manufacturer claimed efficacy after one dose. Without exception, all vaccines required a booster five months after the last dose or two months after the last dose for Ad26.COV2.S [2].

By September 2021, the number of people vaccinated with the full dose schedule was 19,404,212, representing 51.1% of the Polish population. During the same period, the number of people vaccinated with at least one dose was 19,813,605 people, representing 52.2% of the Polish population. By the same time, there had been 2.9 million confirmed cases of SARS-Cov-2 infection and 75,650 associated deaths in Poland [1]. The BNT162b2 vaccine accounted for 78.3% of all COVID-19 vaccine doses in Poland [3].

The COVID-19 vaccination administered from the beginning of 2021 is currently being studied worldwide for its impact on various disease entities. Cases of ocular side effects indicating a direct link to COVID-19 vaccination have been described. Among the complications listed were anterior uveitis, panuveitis, bilateral multifocal choroiditis, central serous retinopathy, graft rejection after Descemet’s membrane endothelial keratoplasty, acute macular neuroretinopathy, and central retinal vein occlusion [4,5,6,7,8,9,10,11,12,13]. Moreover, cases of series of subretinal hemorrhages in the course of neovascular AMD during anti-vascular endothelial growth factor (anti-VEGF) therapy have been observed after receiving the BNT162b2 or ChAdOx1-S vaccine [14].

One of the first patient target groups for vaccination in Poland was the population of people > 60 years of age. In the population above this age, age-related macular degeneration (AMD) is found in more than 25% of European patients [15].

The pathomechanism of AMD is not fully understood, and many factors are involved, including inflammation. In Poland, patients with an exudative form of AMD have been enrolled for treatment under the Polish National Health Fund therapeutic program (TP) since 2015. Treatment in the TP has consisted of intravitreal injections of anti-VEGF agents—0.5 mg of ranibizumab or 2 mg of aflibercept—which is the gold standard, having proven clinical efficacy with a low rate of significant side effects. These substances inhibit vascular endothelial growth factor (VEGF), which is responsible for angiogenesis, and consequently limit the process of neovascularization [16,17].

In response to patients’ concerns and considering the inflammatory pathomechanism of AMD, we evaluated whether COVID-19 vaccination affects the activity of the exudative form of age-related macular degeneration during anti-VEGF therapy. To the best of our knowledge, this is the first article on the subject.

## 2. Materials and Methods

### 2.1. Study Population

This retrospective comparative study included patients with an exudative form of AMD that had been treated with anti-VEGF therapy—aflibercept and ranibizumab—under the TP between December 2020 and September 2021 in the Department of Ophthalmology, Kielce, Poland. The study adheres to the tenets of the Declaration of Helsinki and received approval from the Ethics Committee of Jan Kochanowski University in Kielce (study code 80, approved 30 November 2021).

The inclusion criteria for the study were: treatment with aflibercept or ranibizumab according to the “pro re nata” regimen in the course of neovascular AMD, and “full vaccination” against COVID-19 in the course of AMD treatment. We understand “full vaccination” as receiving one dose of the Ad26.COV2.S vaccine or two doses of BNT162b2, mRNA-1273, or ChAdOx1-S vaccines. Based on inclusion criteria, 158 patients who visited our clinic between December 2020 and March 2021 were reviewed.

Patients were excluded from the study if the loading phase with aflibercept or ranibizumab was not completed at two visits before the first dose of the vaccine, if the anti-VEGF agent was changed, and if they did not receive three injections of the new drug at two visits before the first dose of the vaccine. Moreover, the exclusion criteria included: anti-VEGF agent switch during the follow-up, cataract surgery during the follow-up, postponement of the appointments, eyes with progressive high (degenerative) myopia (the International Classification of Diseases 10th Revision (ICD-10) code H44.2), history of retinal vein occlusion (ICD-10 code H34.8), retinal artery occlusion (ICD-10 code H34.1), residual stage of open-angle glaucoma (ICD-10 code H40.15), optic neuropathy (ICD-10 code H47 and H48), retinal dystrophy (ICD-10 code H35.5), macular holes and telangiectasias (ICD-10 code H35.8), history of retinal detachment (ICD-10 code H33), endophthalmitis (ICD-10 code H44.0), visually significant cataract (ICD-10 code H25 excluding H25.0), previous ocular surgery (besides uncomplicated cataract surgery and intravitreal injections of anti-VEGF agents), photodynamic therapy, macular laser therapy of any kind, media opacity affecting OCT scan or image quality (score less than 8/10), and a history of COVID-19 infection. We also did not include patients vaccinated with Ad26.COV2.S, mRNA-1273, and ChAdOx1-S, in the final study due to the small number of patients.

### 2.2. Data Collection and Study Design

In total, 63 eyes of 59 patients with AMD were analyzed. Follow-up visits typically included best-corrected visual acuity (BCVA) measured by ETDRS chart, slit-lamp examination, dilated fundus examination, and spectral-domain optical coherence tomography imaging (SD-OCT, Copernicus, Optopol Technologies, Zawiercie, Poland) at each visit. Data on the timing and type of vaccine administered were collected based on previous forms routinely collected from patients.

During the process of treatment with anti-VEGF therapy—aflibercept or ranibizumab—patients received two doses of the BNT162b2 vaccine. The records of two previous visits before the first dose of vaccination (first pre-vaccination visit marked as “V-1”, the previous pre-vaccination visit marked as “V-2”), and two subsequent visits after the second dose of vaccination (first visit after the second dose of the vaccine marked as “V1”, second visit after the second dose of the vaccine marked as “V2”) were collected. The SD-OCT test was performed through a dilated pupil, centered on the fovea, and carefully analyzed by the same observer (B.P.-A.).

Structural OCT macular parameters were measured using the early-treatment diabetic retinopathy (ETDRS) grid centered in the fovea by manual fixation. Average central retinal thickness was measured automatically by the SD-OCT software in the central 6 × 6 mm. In addition, the maximum height of each of the following was manually measured using the caliper scale provided by the SD-OCT software: retinal thickness (thRetinal) in the thickest cross-section of the retina, subretinal fluid thickness (thSRF) in the thickest cross-section of the subretinal fluid (SRF), subretinal complex thickness (thSubretinal) in the thickest cross-section of the subretinal complex, total thickness (thTotal) in the maximum thickness from the inner retinal surface to the inner border of the retinal pigment epithelium (RPE). The presence or absence of each of the following was analyzed: intraretinal cystoid fluid (IRC), subretinal fluid (SRF), serous PED (sub-RPE fluid), subretinal hyperreflective material (SHRM), and retinal hemorrhage.

### 2.3. Statistical Analysis

The normality of variable distribution was assessed with the use of the Shapiro–Wilk test. Thus, the nonparametric tests were used. For comparing changes, Wilcoxon’s test was used for dependent pairs of variables, and Friedman’s ANOVA test was used for multiple dependent variables; for comparing changes depending on treatment, U Mann–Whitney test for independent pairs of variables was used [18,19]. The statistical significance threshold has been established as α = 0.05. The analysis was performed with the use of TIBCO Software Inc. (Palo Alto, CA, USA, 2017) Statistica (data analysis software system), version 13 (http://statistica.io, accessed on 10 January 2022).

## 3. Results

The group consisted of 39 women and 20 men, and the mean age of the study group was 79.47. A total of 63 eyes of 59 patients were included in the study. There were 41 patients treated with aflibercept and 22 with ranibizumab. Additional information on the course of treatment of AMD in the studied patients is presented in Table 1.

### 3.1. Visual Acuity Outcomes

There were no statistically significant differences in a best corrected visual acuity (BCVA) test measured on the ETDRS chart between pre- and post-vaccination visits (Table 2).

### 3.2. OCT Quantitative Parameters

The difference in the average retinal thickness was observed between the last and each other checkpoint of the follow-up for the aflibercept group and in the overall outcome (*p* < 0.001). The results are presented in Table 3, Table 4 and Table 5.

The maximum thickness from the inner retinal surface to the inner border of RPE decreased during the observation; differences were observed comparing visits −2 and 1 (459.27 ± 112.88 vs. 443.76 ± 110.05, *p* = 0.0389).

Subretinal complex thickness decreased during follow-up, and the differences were observed between visits −2 and 2 (229.30 ± 115.51 vs. 219.17 ± 119.75, *p* = 0.0276).

No statistically significant difference was observed in the manually measured thickest cross-section of the retina and subretinal fluid thickness (*p* > 0.05).

### 3.3. OCT Qualitative Parameters

No statistically significant difference was observed in the occurrence of intraretinal cystoid fluid, serous PED, subretinal hyperreflective material, and retinal hemorrhage (*p* > 0.05).

### 3.4. Analysis of Time Interval since Vaccination, Dose Number, and Type of Anti-VEGF Agent

There were no differences in the analyzed parameters depending on the time from vaccination to the study, taking into account independently the times for the first and second doses (*p* > 0.05). No differences were found in the analyzed parameters depending on the number of doses administered (*p* > 0.05).

The difference in frequency of occurrence of SRF between aflibercept and ranibizumab was observed in 1 and 2 checkpoints (*p* < 0.05). For the others mentioned, differences were also observed before vaccination, which means that vaccination did not influence the emergence of differences in outcomes according to the drug used (*p* > 0.05).

## 4. Discussion

To our best knowledge, the effect of the BNT162b2 vaccine on AMD activity or the efficacy of anti-VEGF treatment is not yet known.

The involvement of different elements of the immune system in the pathogenesis of AMD has been demonstrated by a number of factors. Previous studies have shown an increased number of inflammatory markers in serum, aqueous humor, or macular tissue in patients with AMD [20,21,22,23,24]. During the course of the disease, RPE cells cannot provide adequate retinal homeostasis, leading to the accumulation of an extracellular material between the RPE and Bruch’s membrane. Disrupted epithelial cells induce an inflammatory response and activate the complement cascade resulting in drusen biogenesis [25,26]. Moreover, RPE cells can damage themselves through a feedback loop by releasing too much of certain pro-inflammatory cytokine [27].

The intended consequence of the mRNA vaccine is to trigger an immune response, leading to host defense, which can promote inflammatory diseases [28]. Studies have shown that the BNT162b2 vaccine triggers a mild innate immune response in the host after the first immunization, increasing after the second immunization [29]. Although there are undoubtedly benefits of vaccination significantly outweighing the rare adverse reactions following vaccination, cases of “over-reaction” of the immune system following COVID-19 vaccination have been reported in the literature. Hyperinflammation has been manifested as adult-onset Still’s disease, macrophage activation syndrome, subacute thyroiditis, acute demyelinating of the central nervous system, orbital inflammatory disease, and autoimmune hepatitis [30,31,32,33,34,35,36,37,38,39,40,41,42].

Considering the immunological pathomechanism of AMD and the possible pro-inflammatory effect of mRNA vaccines, we investigated whether the administration of the mRNA vaccination has an effect on AMD activity.

The main result of the study was a decrease in average retinal thickness, subretinal complex thickness, and total thickness during follow-up towards physiological values, the intended effect after anti-VEGF treatment. Furthermore, we did not observe an increase in the features of the disease activity, based on the occurrence of the features such as IRC, SRF, sub-RPE fluid, and retinal hemorrhages. Moreover, there was no statistically significant difference in the BCVA between pre- and post-vaccination visits, which translates into no functional visual impairment.

There was no comparative data available on the anatomical and visual acuity outcomes in a group of patients at different stages of treatment, as during our study. However, two studies reported visual outcomes with non-naive eyes. Chandra et al. investigated visual acuity outcomes of aflibercept therapy for neovascular age-related macular degeneration for five years and evaluated the outcomes from years two to five [43]. The patients gained +6.3 letters in the first year and +3.9 in the second year. Yang et al. evaluated visual and anatomical outcomes during treatment with ranibizumab [44]. After 12 months of treatment according to the pro-renata regimen, the mean BCVA was 57.9 (20.0) ETDRS letters, and the mean (SD) central subfield retinal thickness (CSRT) was 246.2 (50.8) μm. During 12 months they reported a VA loss of 0.2 ETDRS letters and an increase in CSRT of 6.6 µm. In our study, the mean BCVA during the baseline visit was 60.9 and 57.38 letters in the aflibercept and ranibizumab groups, respectively. The average retinal thickness decreased by 9.22 µm between visits −2 and 2 and the patients gained 0.49 letters in the aflibercept group and lost 1.09 letters in the ranibizumab group, indicating that VA and retinal thickness remained stable for this period.

Park et al. described 11 patients with neovascular AMD and a history of anti-VEGF treatment, who experienced an exacerbation of the disease due to submacular hemorrhage after the BNT162b2 and ChAdOx1 vaccine [14]. Eight patients developed a subretinal hemorrhage after the first dose of the vaccine and three patients after the second dose. The median time since vaccination was two days. Among our patient group, we did not observe macular hemorrhages; however, we observed one case of an extensive submacular hemorrhage after the second dose of the mRNA-1273 vaccine before the start of our study.

Our study has several limitations. Firstly, because the elderly were mainly vaccinated with BNT162b2, we did not include other preparations. In addition, there are not many publications describing the results of anti-VEGF injections in groups of patients at different stages of the treatment process, in particular the anatomical parameters of the retina, so the features of disease activity cannot be accurately compared.

A strength of the study is the relatively large patient group despite the restrictive exclusion criteria. With the availability of pre- and post-vaccination data, the absence of an increase in disease activity can be clearly established. As this is the first study to evaluate the effect of COVID-19 vaccination on AMD activity, and this issue needs further investigation. We intend to expand our study and include patients receiving subsequent vaccination doses, as well as involve other COVID-19 vaccine preparations. We encourage researchers to collaborate on a multi-center study based on a similar methodology.

## 5. Conclusions

In the present study, anti-VEGF treatment resulted in a decrease in retinal thickness, total thickness, and subretinal complex thickness despite vaccination with BNT162b2. We showed that the activity of the exudative form of AMD did not deteriorate after the administration of the vaccination. This issue needs multi-center investigation in a larger group of patients to confirm these findings.

## Figures and Tables

**Table 1 vaccines-10-01878-t001:** Number of anti-VEGF injections at the time of visit “−2”.

Number of Injections	Currently Administered Preparation	Total
Average	9.5	13.7
Minimum	4	4
Maximum	21	32

**Table 2 vaccines-10-01878-t002:** Visual acuity (ETDRS chart) during follow-up.

Treatment	Time of Examination				*p*-Value
	V-2	V-1	V1	V2	
Aflibercept (X ± SD) (*n* = 41)	59.36 ± 12	60.25 ± 12.75	61.16 ± 10.46	68.4 ± 45.98	0.7542
Ranibizumab (X ± SD) (*n* = 22)	57.38 ± 7.68	58.81 ± 8.5	56.43 ± 5.95	55.1 ± 7.92	0.1376
Overall (X ± SD) (*n* = 63)	58.71 ± 10.75	59.78 ± 11.48	59.61 ± 9.44	64.03 ± 38.33	0.3705

**Table 3 vaccines-10-01878-t003:** Average central retinal thickness [μm] during follow-up.

Treatment	Time of Examination				*p*-Value
	V-2	V-1	V1	V2	
Aflibercept (X ± SD) (*n* = 41)	276.27 ± 19.21	274.83 ± 19.35	278.2 ± 28.39	269.24 ± 27.64	**0.0009**
Ranibizumab (X ± SD) (*n* = 22)	266.14 ± 20.57	262.86 ± 22.9	260.68 ± 21.81	252.82 ± 32.53	0.1629
Overall (X ± SD) (*n* = 63)	272.73 ± 20.12	270.65 ± 21.27	272.08 ± 27.42	263.51 ± 30/22	**<0.0001**

**Table 4 vaccines-10-01878-t004:** Post-hoc analysis of average central retinal thickness during follow-up of aflibercept comparisons in pairs).

	V-2	V-1	V1	V2
V-2		0.3442	0.8560	**0.004**
V-1	0.3442		0.3365	**0.0045**
V1	0.8560	0.3365		**0.0001**
V2	**0.004**	**0.0045**	**0.0001**	

**Table 5 vaccines-10-01878-t005:** Post-hoc analysis of average central retinal thickness during follow-up of ranibizumab (comparisons in pairs).

	V-2	V-1	V1	V2
V-2		0.2891	0.0766	**0.0074**
V-1	0.2891		0.7369	0.0702
V1	0.0766	0.7369		0.0716
V2	**0.0074**	0.0702	0.0716	

## Data Availability

All data are available from the first author—B.P.-A.

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
