# Peer review of "BNT162b2 (Pfizer/BioNTech) COVID-19 Vaccination Was Not Associated with the Progression of Activity of the Exudative Form of Age-Related Macular Degeneration during Anti-VEGF Therapy"

_vaccines, 2022, doi:10.3390/vaccines10111878_

Round 1

Reviewer 1 Report

I read it with great interest, but I have raised several concerns.

#1. The study sample is too small to draw the conclusion.

#2. Single-hospital based dataset can not guarantee the generalization.

Reviewer 2 Report

The manuscript reports about the evaluation of activity of exudative AMD during anti-VEGF therapy, i.e. before and after administration of BNT162b2 (Pfizer–14 BioNTech) vaccination.

Anti-VEGF activity was not influenced by administration of Pfizer BioNTech vaccine and exudative AMD was not exacerbated by vaccination.

This comes to little or no surprise, since in the available literature reports are scarce, and among ocular side effects, retinal bleeding has been reported as a sign of central or branch vein occlusion.

Results from this study may help clinicians in reassuring patients that sometimes may appear anxious and worried to receive Covid-19 vaccine during treatment with anti-VEGF for neovascular AMD

Reviewer 3 Report

Dear authors,

I have now completed the review of the manuscript titled "Evaluation of the effect of administration of COVID-19 vaccination with BNT162b2 (Pfizer–BioNTech) on the activity of the exudative form of age-related macular degeneration during anti-VEGF therapy."

In the present study, the authors evaluated the activity of the exudative form of age-related macular degeneration (AMD) during anti-VEGF therapy before and after the administration of BNT162b2 (Pfizer–BioNTech) vaccination.

The manuscript is interesting and, in general, fair written.

I have some minor suggestions to further improve the quality of the manuscript.

1.  Ln 66-71: The authors claimed that the data source of this study is from the retrospective cohort of AMD that have been treated with aflibercept and ranibizumab under the Polish National Health Fund therapeutic program (TP) between December 2020 and September 2021. To increase the accessibility of the data and clarity of this study, I suggest authors clarify how other researchers can obtain the original data, i.e., the Polish National Health Fund therapeutic program dataset.

2. Ln 84-86: Several diagnosis criteria has made here. myopia, RVO, RAO, glaucoma, optic neuropathy, and so on... How did the authors define diagnosis? If you used ICD-9 or ICD-10 codes, please clarify all. If you used a questionnaire, attach it to the supplement section.

3. Ln 122-128: In the ‘Statistical analysis’ section, the paragraph has no references for selecting your method. Please justify yourself about selecting those methods, or refer to some articles on how to select proper statistical methods. For example:

~~ were used while comparing changes depending on treatment U Mann-Whitney test for independent pairs of variables was used[add a reference related to methods for testing statistical differences between groups].

4. Ln 220-227: What is the future scope of the proposed research, authors have described the limitations in a good way, and I suggest that these can be the future scope of the work. The further study suggestion would be helpful to readers who are interested in this study. The fellow researchers can do subsequent research or similar research topic as per your suggestions.

5. The manuscript will be much better if an English proofing service is used.

Reviewer 4 Report

I have gone through the manuscript title "Evaluation of the effect of administration of COVID-19 vaccination with BNT162b2 (Pfizer–BioNTech) on the activity of the exudative form of age-related macular degeneration during anti-VEGF therapy. I have some suggestions to enhance the quality and readability of the present manuscript. It is a nice study, authors have tried to observe the effect of COVID-19 vaccination has not yet been described in patients with AMD treated with intravitreal injections of anti—VEGF agents.

1-The title of the manuscript needs to be more focused and clearer.

2- The introduction needs to focus on the literature background AMD related to the earlier study and the novelty of the study need to describe in detail

3- This is indeed a nice case study on AMD but needs more detailed discussion.

4-Conclusion needs to be improved.

Round 2

Reviewer 1 Report

In order to publish this manuscript, the authors need an appropriate answer to my comments.

#1. Title: effect -> The word "effect" is not available in observational studies. Please revise it (effect -> association).

#2. Comparing changes in the Wilcoxon’s test for dependent pairs of variables and Friedman’s ANOVA test for multiple dependent variables were used while comparing changes depending on treatment U Mann-Whitney test 133for independent pairs of variables was used. -> Please cite the statistical guidelines additionally such as DOI: https://doi.org/10.54724/lc.2022.e1

#3. P value 0,7542 -> You have to write a dot, not a comma.

#4. If the Editor decided to publish, This could consider publishing.

Author Response

Reviewer:

In order to publish this manuscript, the authors need an appropriate answer to my comments.

#1. Title: effect -> The word "effect" is not available in observational studies. Please revise it (effect -> association).

#2. Comparing changes in the Wilcoxon’s test for dependent pairs of variables and Friedman’s ANOVA test for multiple dependent variables were used while comparing changes depending on treatment U Mann-Whitney test 133for independent pairs of variables was used. -> Please cite the statistical guidelines additionally such as DOI: https://doi.org/10.54724/lc.2022.e1

#3. P value 0,7542 -> You have to write a dot, not a comma.

#4. If the Editor decided to publish, This could consider publishing.

Answer to the review, not included in the text:

We would like to express our gratitude for taking the time to review our article. All suggestions were very helpful and we agree with all of them, which has been included in the updated version of the manuscript. We also ensure that the manuscript is sent to Language Editing Service. 

Reviewer:

#1. Title: effect -> The word "effect" is not available in observational studies. Please revise it (effect -> association).

Additional information included in the manuscript:

Title:

Association of BNT162b2 (Pfizer–BioNTech) COVID-19 vaccination with the activity of the exudative form of age-related macular degeneration during anti-VEGF therapy

Reviewer:

#2. Comparing changes in the Wilcoxon’s test for dependent pairs of variables and Friedman’s ANOVA test for multiple dependent variables were used while comparing changes depending on treatment U Mann-Whitney test 133for independent pairs of variables was used. -> Please cite the statistical guidelines additionally such as DOI: https://doi.org/10.54724/lc.2022.e1

Additional information included in the manuscript:

References:

[19] Lee SW. Methods for testing statistical differences between groups in medical research: statistical standard and guideline of Life Cycle Committee. Life Cycle 2022;2:e1

Reviewer:

#3. P value 0,7542 -> You have to write a dot, not a comma.

Answer to the review, not included in the text:

We have changed all the commas to dots in the numbers.